# Evaluating Self-Directed Behaviours and Their Association with Emotional Arousal across Two Cognitive Tasks in Bonobos (*Pan paniscus*)

**DOI:** 10.3390/ani12213002

**Published:** 2022-11-01

**Authors:** Daan W. Laméris, Jonas Verspeek, Marina Salas, Nicky Staes, Jonas R. R. Torfs, Marcel Eens, Jeroen M. G. Stevens

**Affiliations:** 1Behavioural Ecology and Ecophysiology Group, Department of Biology, University of Antwerp, 2610 Wilrijk, Belgium; 2Antwerp ZOO Centre for Research & Conservation (CRC), Royal Zoological Society of Antwerp (RZSA), 2018 Antwerp, Belgium; 3SALTO, Agro- and Biotechnology, Odisee University College, 1000 Brussels, Belgium

**Keywords:** arousal, bonobo, displacement behaviour, emotion, great ape, laterality, self-directed behaviour, touchscreen

## Abstract

**Simple Summary:**

Self-directed behaviours (SDBs), such as self-scratching or self-touching, are commonly used as indicators of stress or poor welfare in animals. However, whether these behaviours truly reflect stress may depend on individual behaviour, species, context, and to which side of the body they are directed. Namely, one idea is that negative emotions are processed more frequently in the right brain, and because these nerves end in the opposite side, the following sensation is experienced in the left side of the body. Not much is known about the reliability of SDBs as indicators of stress in bonobos. Therefore, we investigated the production and asymmetry of SDBs in bonobos whilst they completed two cognitive touchscreen tasks. The most common SDB was nose wiping, followed by gentle self-scratching, then rough self-scratching. When the bonobos made incorrect responses, due to their unsuccessful experience resulting in expressions of frustration, they showed more nose wiping and rough self-scratching. Additionally, rough self-scratching was more directed to the left side of the body, suggesting a link to negative emotions. Interestingly, in one of the tasks, the bonobos gently self-scratched more frequently when they gave correct responses, possibly indicating positive emotions. These results increase our understanding of SDBs as indicators of emotion in bonobos.

**Abstract:**

Self-directed behaviours (SDBs) are widely used as markers of emotional arousal in primates, and are commonly linked to negative arousal, or are used as indicators of stress or poor welfare. However, recent studies suggest that not all SDBs have the same function. Moreover, lateralisation in the production of these behaviours has been suggested to be associated with emotional processing. Hence, a better understanding of the production and the asymmetry of these displacement behaviours is needed in a wider range of species in order to confirm their reliability as indicators of emotional arousal. In the current study, we experimentally evaluated the production and asymmetry of SDBs in zoo-housed bonobos during two cognitive touchscreen tasks. Overall, nose wipes were most commonly observed, followed by gentle self-scratches, and rough self-scratches. The rates of nose wipes and rough self-scratches increased with incorrect responses, suggesting that these behaviours indicate arousal and possibly frustration. Rough self-scratching was additionally more directed towards the left hemispace after incorrect responses. In contrast, gentle self-scratching increased after correct responses in one study, possibly linking it with positive arousal. We also tested if left-handed bonobos showed greater behavioural reactivity towards incorrect responses, but found no evidence to confirm this hypothesis. Our results shed light on potential different mechanisms behind separate SDBs. We therefore provide nuance to the use of SDBs as indicator of emotional arousal in bonobos.

## 1. Introduction

Self-directed behaviours (SDBs), behaviours that are directed at an animal’s own body, are considered to be displacement behaviours that result from frustration and/or internal conflict within an animal [1,2]. In non-human primates, SDBs, such as self-scratching or self-touching, have been introduced as behavioural indicators of psychosocial stress [3], because (1) pharmacological evidence with anxiogenic and anxiolytic drugs provided support for the link between SDBs and social anxiety, arousal or stress [4,5]; (2) observational studies indicated increased rates of SDBs with social or environmental stressors [3,6,7,8,9]; and (3) SDBs have been shown to decrease after positive affiliative interactions [10,11] or after reconciliation following agonistic interactions [12,13]. Other studies found that SDB rates increased with the complexity of cognitive tasks (chimpanzees (*Pan troglodytes*): [14]); when delay between trials increased (orangutan (*Pongo pygmaeus*): [15]); and when subjects made incorrect responses (chimpanzees: [16,17]; mandrills (*Mandrillus sphinx*): [18]), suggesting that SDBs also reveal emotional arousal in non-social contexts when the subjects are not achieving their goals [1].

Due to the rare occurrence of SDBs, and because the differences between them are subtle, many studies combine different SDB types, for example, gentle and rough self-scratching, or combine SBDs with other behaviours, such as self-grooming or self-plucking, into one measure [13,19], which obscures their interpretation. On the other hand, several studies have indicated subtle nuances between different SDBs [18], or between individual differences in the rates of SDBs [17]. For example, some studies suggest that in chimpanzees, rough self-scratches, but not gentle self-scratches or self-grooming, indicate negative arousal [6,20,21]. Others have suggested that gentle self-scratching may reflect lower levels of negative arousal [17]. One particular SDB in great apes, that is often overlooked despite being commonly observed, is ‘nose wiping’ [22,23]. This behaviour is rather inconspicuous, and different studies have referred to this behaviour with different terms, e.g., ‘nose gesture’ [17] or ‘rubbing’ [24]. In chimpanzees, there is some evidence that nose wipes appear to increase with errors in cognitive tasks [17], but not with cognitive challenge [25].

To better understand the potential link between SDBs and (negative) affect, researchers have also focused on the asymmetrical production of these behaviours [14,21,26]. Lateralised behaviours are associated with specialisation of the left or right brain hemispheres, which results in the differential processing of information, perception and production of emotions across vertebrates [27]. Although different views exist [28], findings largely suggest a similar pattern of a right hemisphere bias for expressing intense emotions [29], specifically negative emotions, such as stress [30]. Since SDBs are typically executed using one hand, they represent a lateralised behaviour, and looking at such biased production presents a potential key variable in identifying the link between these behaviours and their emotional valence, i.e., whether the SDB reflects positive or negative emotions. Observational studies yield inconsistent evidence for the asymmetrical production of SDBs in great apes. A left-hand bias for face touching has been observed across orangutans, gorillas (*Gorilla gorilla*) and chimpanzees [31], whereas a right-hand bias was found for self-scratching in chimpanzees [24]. Other studies found no overall hand preference for SDBs in these species [22,24,32]. Within the context of measuring arousal in cognitive challenges, studies have reported more right-hand SDBs when the chimpanzees made errors on the task [14,21], although a more recent study found that chimpanzees and gorillas had a left-hand bias for SDBs during incorrect trials [26].

In addition to hand preference for SDBs, changes in the target location on the body (i.e., left or right hemispace) have been associated with asymmetrical processing of emotions in the brain. Chimpanzees and gorillas direct self-scratches more to the left side of their body, supporting the view of right hemispheric processing [24,26], whereas another study found that in a chimpanzee, rubs were more directed to the right hemispace after incorrect responses, while self-scratches (both gentle and rough) were more directed to the left hemispace [21].

Altogether, current evidence suggests that SDBs in primates may reflect arousal, and possibly negative arousal, but that their reliability as an indicator for concepts, such as stress, frustration, or anxiety, may depend on factors such as species, context, SDB type, hand use, and target location. The purpose of this study is to increase our understanding regarding the production of SDBs in great apes, and more specifically, in bonobos (*Pan paniscus*). Bonobos represent an interesting study species for an examination of frequency and asymmetry of SDBs during cognitive tasks for several reasons. Firstly, chimpanzees and bonobos appear to differ in handedness. Several studies indicate that, whereas chimpanzee populations show right-handed bias in gestural communication, unimanual reaching, and bimanual complex coordination, bonobos have individual preferences for left- or right- handedness, but, besides one study [33], no clear right-hand bias across populations in various contexts [34,35,36,37,38]. Nonetheless, greater leftward asymmetries in brain regions associated with the motor skills used for manual actions have been observed [39]. Second, chimpanzees and bonobos may differ in their emotional decision making [40], and studies of brain regions have indeed identified differences in neural systems that regulate emotional processing, such as the amygdala [41,42,43]. Therefore, investigating bonobos’ asymmetry in SDB production can further shed light on the different mechanisms behind these behaviours across species. In addition, a better understanding of the contexts of SDB production and how they relate to (negative) emotional arousal has implications for the use of these behaviours in assessment of affective states and welfare.

In this study, we examine the production of four SDBs (nose wiping, gentle self-scratching, rough self-scratching, and self-touching) in bonobos during two cognitive touchscreen tasks and evaluate the asymmetry of their production in relation to trial accuracy. We expect to find that (1) some, but not all, SDBs will increase with errors made during the tasks; (2) if SDBs are a reflection of internal arousal, they are produced more with the left hand; (3) SDBs that are linked with arousal are targeted more to the left hemispace of the body after incorrect responses, compared to those associated with correct responses; (4) that left-handed individuals show enhanced behavioural reactivity, and thus more SDBs in response to incorrect answers.

## 2. Materials and Methods

### 2.1. Subjects and Housing

The study subjects were eight mother-reared adolescent and adult bonobos (three females and five males; mean age = 15.8 years, range = 7–27 years; Table 1) who were part of a social group of 20 individuals, housed at Zoo Planckendael (Belgium). The bonobos were housed in an indoor enclosure (total surface 422 m^2^) consisting of ten interconnected rooms, of which four main rooms were visible for zoo visitors, and six rooms off exhibit. When the temperature allowed, the bonobos had access to an outdoor enclosure (3000 m^2^). Fresh vegetables, fruits, browse, and primate chow was provided four times per day and the bonobos had access to water ad libitum.

### 2.2. Testing Procedure

Touchscreen sessions took place four to five times per week, between 12:00 and 15:00, in the off-exhibit enclosures. Subjects could choose to participate voluntarily in touchscreen sessions and were not separated from group members for testing.

All sessions were conducted on a 22’ Viewsonic TD2220 touch-sensitive monitor (1920 × 1080 resolution), which was connected to the researcher’s (DWL) computer. A second monitor allowed the researcher to view the subject’s responses. The touchscreen setup was mounted on an adjustable cart, placed outside an off-exhibit enclosure. The touchscreen was placed parallel to the enclosure mesh, allowing the bonobos to work on the touchscreen through the mesh. Training and testing tasks were designed using OpenSesame [44]. Stimulus preparation was conducted in Adobe Photoshop version 21.2.2.

The apes were rewarded for correct responses with an automatic delivery of a DK Zoological Trainings Biscuit (small), triggered by a custom-made pellet dispenser. A secondary reinforcing tone was played via two speakers behind the touchscreen. Primary and secondary reinforcers were delivered on a 100% fixed reinforcement ratio. Additionally, we manually provided a raisin, through a PVC tube, on every fifth correct response to maintain the bonobo’s interest. If an individual finished all of the trials within a session, they received three peanuts. Each response was followed by a 1500 ms inter-trial interval (ITI). When a bonobo made an incorrect response, no reinforcement was provided, and the ITI was increased to 3500 ms.

### 2.3. Touchscreen Tasks

This study reports observations regarding the production of SDBs across two studies. Eight individuals participated in Study 1, and a subset of four individuals participated in Study 2.

Study 1 was a response-slowing task [45], conducted between January and July 2021. In this task, the bonobos were trained to touch grey square target stimuli. During the period of this study, the bonobos were housed in two sub-groups, the composition of which was regularly changed, to mimic natural fission-fusion dynamics. Test sessions for this study were conducted during a pre-baseline, on days with fission-fusion events, and during post-baseline days, one day after the fission-fusion events. The stimuli in the test sessions included images of a frontal bonobo face picture with a neutral expression (i.e., a direct gaze), or a profile view bonobo face picture with a neutral expression (i.e., an averted gaze), see Figure 1A. The maximum number of trials per subject, per day, was set to 60.

Study 2 was a pictorial emotional Stroop task [46], conducted between September and October 2020 [47]. Prior to the testing sessions, the subjects successfully completed colour-discrimination training, which was required for participation. The detailed protocol is described in [47]. In short, the bonobos were trained to always touch the stimuli that were framed in a target colour, while an identical stimulus was simultaneously presented and framed in a different distractor colour. Hence, while stimuli were similar, the bonobos could make a correct response (i.e., touching the stimulus with the target colour) or an incorrect response (i.e., touching the stimulus with the distractor colour). The study itself consisted of three parts: (1) a colour-interference Stroop task, in which bonobos were shown geometric shapes that were either the same colour as the frame or a different colour; (2) a social, pictorial emotional Stroop task, in which bonobos were shown images of unfamiliar bonobos that had different facial expressions that are typically expressed in negative, neutral or positive contexts; (3) a non-social, pictorial emotional Stroop task, in which bonobos were shown biologically relevant objects that were predicted to have a negative (i.e., a leopard), neutral (i.e., a flower), or positive (i.e., a highly preferred food item) association, see Figure 1B. The maximum number of trials per subject per day was set to 105.

### 2.4. Video Coding

All test sessions were video-recorded using a Canon Legria HF R88. We followed the same coding protocol for both studies and coded the following factors: (a) hand used to complete the touchscreen trial (left or right); (b) any and all SDBs; (c) hand used to perform the SDB (left or right); (d) hemispace to which the SDB was directed (left, mid, or right). The software automatically recorded the accuracy (i.e., correct or incorrect response), and the start and end time for each trial.

Based on previous studies on chimpanzees and bonobos [14,17,18,23,26], we identified 4 SDBs: nose wipes, gentle self-scratches, rough self-scratches, and self-touches. Nose wipes include when the subject raises the arm with a relaxed hanging hand and moves the wrist or back of the hand downwards across the nose [23]. Gentle self-scratches were defined as a subject raking their own hair or skin with bent fingers [6]. Rough self-scratches refer to the raking of one’s own hair or skin with fingernails, including large movements of the arm [6]. A new gentle or rough scratch event was recorded after a period of five seconds without the corresponding scratching behaviour, or if the location of the scratch changed. A self-touch was defined as a single moment of contact between the fingers and another body part, without raking motions. We included self-touch as a separate category, as we considered this different to self-scratches. Examples of SDBs are presented in Appendix A.

To test the reliability of the coding, 16% of the trials were coded by two observers, who were blind to the study aims. The reliability of: hand used to complete the touchscreen, hand used to perform SDBs, and to which hemispace the SDB was directed, were assessed using Cohen’s Kappa, and intraclass correlation for the occurrence of the four SDBs using a two-way mixed models with a consistency definition. Inter-observer reliability for hand use on the touchscreen was perfect (=1.00), and almost perfect for hand use to perform SDBs (=0.94) and hemispace (=0.90). Intraclass correlation was moderate for self-touches (=0.70), good for rough self-scratching (=0.80), and excellent for gentle self-scratching (=0.96) and nose wiping (=0.94).

### 2.5. Tube Task

One of our aims was to examine if left-handed individuals show greater behavioural reactivity towards incorrect responses. One obvious way to determine handedness is to look at which hand is used to complete the touchscreen task. However, different levels of manual lateralisation are expected based on the complexity of the task; low-level tasks may reveal a hand preference that is not indicative of the specialisation of the contralateral hemisphere [48]. The touchscreen task may represent such a low-level unimanual task. To obtain a more reliable level of hemispheric specialisation, we therefore completed the ‘tube task’ [49]. We provided the bonobos with PVC tubes probed with small amounts of honey, which encourages the bonobos to hold the tube with the subordinate hand while removing the honey with their dominant hand, therefore presenting a more high-level coordinated bimanual task.

### 2.6. Data Analyses

We analysed Study 1 and Study 2 separately, as they differed in the individuals that participated as well as in task contingency. Before analyses, we excluded outlier trials, i.e., trials where the subject moved out of view, where other bonobos approached and interrupted the subject, or where a behaviour could not be reliably coded (2026 (15.9%) of the trials). Furthermore, we excluded SDBs directed to the mid-line of the face or body from analyses, as no obvious hemispace effect could be assigned (82 (0.6%) of the trials). Self-touches were not analysed due to the low rate of occurrence.

#### 2.6.1. Handedness and Side Index

We used counts of left- and right-handed responses to complete the trials during the touchscreen task (HI-screen) and tube task (HI-tube) to quantify individual hand preferences. The Handedness Index score was computed for each subject as follows:(1)HI=(R−L)(R+L)

Here, R and L correspond to the count of right and left responses.

HI-values range from −1.0 to +1.0, with positive values reflecting greater right-hand use, while negative values indicate more left-hand use. Side Indices (SI) for gentle and rough self-scratching, and HI for performing combined and separate SDBs, were calculated in a similar way. 

Our sample included individuals with varying HI-screen. Therefore, we considered it likely that in-task hand use influences which hand is subsequently used to perform an SDB; we ran a binomial mixed model with subject ID as random intercept to verify this. Indeed, hand use for working on the touchscreen was not independent from the hand use for performing SDBs (χ² = 262.18, df = 1, *p* < 0.001). However, hand use while working on the touchscreens did not predict to which hemispace the SDB would be directed (χ² = 0.054, df = 1, *p* = 0.817).

#### 2.6.2. Linear Mixed Models

We assessed the accuracy of the four individuals that participated in both studies to develop a sense of the perceived difficulty of the two tasks. We applied a generalised linear mixed model (GLMM), with a binomial distribution, with trial accuracy as a dependent variable and study (categorical; Study 1 or Study 2) as a fixed factor. The subject ID was included as a random intercept.

To examine the link of the production of SDBs during the touchscreen sessions and emotional arousal, we created LMMs with SDB rates (per trial per second) as dependent variable for each separate SDB type (nose wipe, gentle self-scratch and rough self-scratch), and trial accuracy (categorical; correct or incorrect) and the hemispace (categorical; left or right side) to which the SDB was directed as independent variables. We included a two-way interaction between trial accuracy and hemispace. As nose wipes are, by definition, directed towards the centre of the face, we only included trial accuracy as a predictor in this model. Subject ID was included as random intercept in all models. We did not include hand use for SDB as an independent variable because this variable was not independent of in-task hand use. We used planned post-hoc testing for significant global effects using simple contrasts, focusing on the effect of trial accuracy. Tukey corrections were applied for multiple comparisons.

To rule out the possibility that condition in Study 1, or stimulus type in Study 1 and Study 2, influenced the production of SDBs, we ran separate LMMs for each SDB, including condition (categorical; pre-baseline, fusion, post-baseline) and stimulus (categorical; direct, averted, control) as the fixed factor for Study 1. For Study 2, we ran separate models for the three experiments, and included stimulus (categorical; Experiment 1—congruent, incongruent, control; Experiment 2—negative, positive, neutral; Experiment 3—negative, positive, neutral) as the fixed factor. Subject ID was again included as a random intercept in all models. Model outputs are presented in Appendix A, and mostly returned insignificant results. Only the rates of gentle self-scratching were lower during trials when averted stimuli were presented, compared to the control trials. Furthermore, the rates of nose wiping were lower during trials with positive social stimuli, compared to the control trials.

#### 2.6.3. Behavioural Reactivity

To examine the difference in behavioural reactivity to incorrect responses, depending on the handedness of the individual, we calculated standardised ratios for each SDB, following previous work [17]. As such, we calculated the average rate of SDBs after incorrect responses and divided this by the average rate after correct responses. This proportion was then standardised. We ran Pearson’s correlations between these standardised ratios and the HI-screen, and against HI-tube.

## 3. Results

### 3.1. Production of SDB Types Per Study

We analysed 10,600 trials (Study 1 = 5876 trials, range = 436–896 per individual; Study 2 = 4724 trials, range = 1134–1220 per individual) and recorded a total of 1537 SDBs. Table 2 presents the distribution of observed SDBs per study. Overall, nose wipes occurred most frequently in both studies, followed by gentle scratching, rough scratching, and self-touching. Individual rates of SDBs were typically low, and are presented in Appendix A.

### 3.2. Accuracy in the Two Studies

To examine the difference in the complexity of the two studies, we only included those individuals that participated in both studies (*n* = 4). The binomial GLMM showed a significant effect of ‘study’ on accuracy scores (χ² = 481.08, df = 1, *p* < 0.001); with accuracy being higher in Study 1 (M = 0.978, SE = 0.006), compared to Study 2 (M = 0.758, SE = 0.003; t_(8065)_ = −21.934, *p* < 0.001). From these results, we concluded that Study 1 was less difficult than Study 2.

### 3.3. Linear Mixed Models

We present HI- and SI-indices in Table 3, from which it can be seen that, based on the HI-screen, our sample of bonobos consisted of two left-, five right-handed individuals, and one ambiguous-handed individual. As the hand used to execute SDBs was not independent of the hand used in the preceding touchscreen task, we decided to focus on laterality effects in the hemispace [21,24]. In order to present a complete perspective, we also present HI-indices for the different SDB types, although it is important to note that these values are dependent on the hand used to complete the touchscreen tasks. These values are presented in Table 3 and revealed that left-, right-, and ambiguous-handed individuals were included in the sample.

#### 3.3.1. Study 1

Eight bonobos participated in Study 1. For nose wiping, we only examined the effect of trial accuracy as this behaviour is directed towards the nose, and therefore hemispace effects are irrelevant. Here, trial accuracy had an effect on nose wiping (χ² = 5.989, df = 1, *p* = 0.014), with rates increasing after incorrect responses compared to correct responses (Figure 2A; t_(5846)_ = 2.447, *p* = 0.014). We did not find a two-way interaction between trial accuracy and hemispace for gentle self-scratching (χ² = 0.102, df = 1, *p* = 0.750), or for rough self-scratching (χ² = 0.146, df = 1, *p* = 0.702). After removing the insignificant interaction effects, we only found a significant main effect of hemispace for rough self-scratching (χ² = 3.866, df = 1, *p* = 0.049) and post-hoc testing showed that rough self-scratches were more directed to the left hemispace (Figure 2B; t_(220)_ = 1.966, *p* = 0.051). Full model results are presented in Table 4, and post-hoc results of the final models are presented in Appendix A.

The participants in Study 2 consisted of a subset of the participants in Study 1, therefore, we ran additional analyses for Study 1, for those four bonobos that participated in both studies. Results were comparable in that we found a significant effect of trial accuracy on nose wiping (χ² = 16.073, df = 1, *p* < 0.001), with higher rates after incorrect responses (t_(3342)_ = 4.009, *p* < 0.001), and no interaction effects between hemispace and trial accuracy for both rough self-scratching (χ² = 1.187, df = 1, *p* = 0.276) and gentle self-scratching (χ² = 1.247, df = 1, *p* = 0.264). In contrast to the analyses on the full dataset, the main effect of hemispace on rough self-scratching was not significant (χ² = 2.614, df = 1, *p* = 0.106). Full post-hoc results of the final models are presented in Appendix A.

#### 3.3.2. Study 2

Four bonobos completed Study 2. Trial accuracy showed a significant effect on nose wiping (χ² = 21.216, df = 1, *p* < 0.001), with higher rates after incorrect responses (Figure 2C; t_(4696)_ = 4.606, *p* < 0.001). Gentle self-scratching was not influenced by the interaction between trial accuracy and hemispace (χ² = 0.031, df = 1, *p* = 0.860), but was influenced by the main effect of trial accuracy (χ² = 8.328, df = 1, *p* = 0.004). Namely, gentle self-scratching increased after correct responses compared to incorrect responses (Figure 2D; t_(119)_ = 2.886, *p* = 0.005). For rough self-scratching, we found a significant two-way interaction between trial accuracy and hemispace (χ² = 6.469, df = 1, *p* = 0.011). The bonobos scratched more to their left hemispace after an incorrect response compared to a correct response (Figure 2E; t_(123)_ = 4.556, *p* < 0.001). This accuracy effect was not observed in the right hemispace (t_(145)_ = 1.311, *p* = 0.192). Using hemispace as a simple contrast, we found that rough self-scratching was more directed to the left compared to the right hemispace during incorrect trials (t_(135)_ = 2.250, *p* = 0.026), but not during correct trials (t_(141)_ = −1.210, *p* = 0.228). Full model results are presented in Table 4, and post-hoc results of the final models are presented in Appendix A.

### 3.4. Handedness and Behavioural Reactivity

No correlation was found between the HI-screen and the behavioural reactivity of nose wiping (Pearson’s r_(6)_ = −0.469, *p* = 0.241), gentle self-scratching (Pearson’s r_(6)_ = 0.146, *p* = 0.731), or rough self-scratching (Pearson’s r_(6)_ = −0.190, *p* = 0.652), or between HI-tube and nose wiping (Pearson’s r_(6)_ = −0.311, *p* = 0.453), gentle self-scratching (Pearson’s r_(6)_ = 0.482, *p* = 0.0.226), or rough self-scratching (Pearson’s r_(6)_ = −0.130, *p* = 0.758).

## 4. Discussion

We studied the production of SDBs in bonobos during two cognitive touchscreen tasks. As expected, and in line with previous studies, we found that bonobos also respond to arousing events with increased rates of some SDBs, namely nose wiping and rough self-scratching. Interestingly, gentle self-scratching increased with correct responses in one study.

Nose wiping was by far the most recorded SDB, constituting 75.5% of all recorded SDBs, followed by gentle self-scratching (18.4%) and rough self-scratching (7.2%). Despite being the most common SDB in our study, not much is known about nose wiping and its potential link to arousal. Some suggest a link to nervousness or edginess [22]. However, with the exception of this report, nose wiping remains overlooked as a possible SDB, and empirical evidence is lacking. We found that rates of nose wiping increased during incorrect trials, and these changes were consistent across the two studies. Similar to other studies, when the bonobos made an incorrect response, they were not given a small food reward and received a short time out, which arguably resulted in increased arousal. A previous study on chimpanzees examined nose wiping and reported changes in rates based on trial accuracy in some subjects, although rates were typically lower than rates of self-scratching [17]. This could hint at species-specific differences in the expressive patterns of SDBs. It is possible that a mutation in the serotonin receptor, linked to increased rates of self-scratching in chimpanzees, is absent in bonobos [50] and could relate to this difference. The fact that nose wiping was so common in the current sample, combined with the observation that rates also increased with incorrect responses in Study 1, which were relatively rare, could suggest that nose wiping is a behavioural response to low levels of arousal in bonobos. Overall, this could support the idea that, compared to chimpanzees, bonobos differ in their behavioural reactivity towards emotional arousal [40]. One other study found that one particular bonobo began nose wiping more after viewing emotional images [51], yet additional exploratory analyses on our data revealed lower rates of nose wiping after viewing play faces. This is interesting as we previously found that these positive social stimuli specifically grab the attention of bonobos [47]. Our results contribute to the limited knowledge regarding nose wiping and suggest that it can be considered an indicator of arousal in bonobos, and potentially of low levels of arousal, although this, and a possible response to emotional stimuli, warrants further validation.

We further aimed to assess handedness for executing SDBs during the cognitive tasks. Based on previous studies in chimpanzees, we expected to find asymmetry in hand use for executing SDBs in bonobos, especially when arousal was increased, e.g., when making incorrect responses [14,24,26]. However, we found that the hand used to perform SDBs was strongly associated with the hand used to complete the trial preceding these SDBs. Whilst changes in hand use can of course still offer information about these asymmetries [26], because our sample consisted of individuals with varying handedness levels, in both touchscreen performance and in the conventional tube task, we reasoned that this would complicate the interpretation of these results. We therefore refrained from testing the effect of hand use for SDBs, but focused on asymmetries in the hemispace (i.e., to which side of the body the SDBs were directed). Since only gentle and rough self-scratches are clearly directed towards one of the two hemispaces, and nose wipes are by definition directed towards the middle of the face, we limited our hemispace analyses to these two behaviours. Results from these analyses only revealed an arousal-related hemispace effect for rough scratching. Namely, when the bonobos made an incorrect response, rough self-scratches were directed to the left hemispace. This suggests that there is a left hemispace bias with increased arousal, which is consistent with previous work on chimpanzees [21,24], and follows the idea of right hemisphere asymmetries for emotional responding [29,30], which then has consequences for asymmetries in cutaneous sensations [14]. However, it is important to note that we only observed this arousal-related hemispace effect in Study 2, while a general left hemispace bias was found in Study 1. This can have multiple explanations. Study 2 was perceived by the bonobos to be more challenging, as indicated by the lower accuracy scores in this task. Baker & Aureli (1997), despite assessing rough self-scratching in different contexts, reported that rough self-scratching may reflect higher levels of anxiety [6], and Troisi et al. (1991) raised the idea that self-scratching and arousal are associated in an inverse-U fashion [52]. Although we were unable to test this, the difference in the effect of trial accuracy on rough self-scratching between the two studies could suggest that the frequency of incorrect responses is a modulating factor in the expression of rough self-scratching. However, one drawback of our study is that Study 1 and Study 2 differed in the number of participants, making it difficult to truly distinguish between the effect of task complexity and the subject sample. Therefore, we re-ran the analysis of Study 1, with only the bonobos that also participated in Study 2, and found that the general hemispace effect on the entire sample disappeared, suggesting that this effect was sample-specific. This is in agreement with an earlier study that focused on individual differences in which subjects are sensitive to incorrect responses [17]. However, due to the relatively low sample size, interpretation of the results should be met with caution.

One individual factor that could explain differences in behavioural reactivity when experiencing arousal is hemispheric specialisation. Several lines of evidence suggest an association with hemispheric specialisation and the stress response [53,54]. We predicted that individuals showing right handedness would show stronger behavioural responses after incorrect trials. We measured handedness during the touchscreen task as a proxy for hemispheric specialisation and tested its effect on behavioural reactivity to incorrect responses. Contrary to our prediction, we found no evidence for an association between handedness and behavioural reactivity. However, hand preference may vary between tasks [55], and it could be that the hand preference measured in the unimanual touchscreen task is not correlated with hemispheric specialisation [48]. For this reason, we additionally correlated behavioural reactivity with incorrect responses to handedness during the tube task, a standardised task to approach hemispheric specialisation in primates [49]. Based on this, we saw that handedness while working on touchscreens was not correlated with handedness measures based on the tube task. This may confirm that the unimanual nature of working on touchscreens taps into different mechanisms than more complex bimanual tasks, such as the tube task, and therefore does not reflect hemispheric specialisation. Nonetheless, handedness based on the tube task also did not show an association with behavioural reactivity during incorrect trials. Evidence on the putative link between handedness and stress response is inconsistent (e.g., higher levels of plasma cortisol were observed in right-handed rhesus macaques [56] and common marmosets [57]), and may not be straightforward. Furthermore, although the increased rates of some SDBs after incorrect responses suggest heightened levels of arousal, it remains unclear which emotions the bonobos were experiencing, and if whether is in fact linked to increased activation of one of the two brain hemispheres.

Interestingly, the bonobos had higher rates of gentle scratching during correct trials in Study 2, compared to incorrect trials. Although this result should be interpreted with caution due to the lower sample size, this finding is consistent with a study on common marmosets (*Callithrix jacchus*), which reported increased rates of self-scratching in positive conditions [58]. Specifically, the authors of the latter study found increased rates during social play, but decreasing rates during food foraging, and no change during food anticipation. The differential patterns of self-scratching in this study highlight the complex nature of self-scratching. We could reason that the increase of gentle self-scratching in our own study may be linked to the anticipation for the food reward. Prior to taking part in these studies, the bonobos were conditioned on an auditory reinforcer, which was accompanied with a small food reward. However, these rewards were automatically triggered and delivered immediately after a bonobo gave a correct response, and we consider it most likely that any scratching occurred after the delivery of these rewards. The timing of food rewards (i.e., immediate, delayed or no reward) in a similar context previously did not affect gentle self-scratching in chimpanzees [21], and we are therefore unsure if the increased rates of gentle self-scratching reflect positive anticipation. Alternatively, because the bonobos participated in the touchscreen sessions in a social setting, it could be possible that they experienced arousal due to competition with other group members when receiving a food reward. However, we trained the bonobos to complete their tasks individually, and paused sessions when they were interrupted by other individuals, in an attempt to avoid competition over the food rewards. The fact that we only observed this effect in Study 2, and not in Study 1 (which was perceived as easier), could suggest that the more frequent incorrect responses enhanced the relative rewarding experience during correct trials, although this is purely speculative. This raises more questions regarding the mechanisms behind this behaviour, and several hypotheses remain to be tested regarding the increase in gentle self-scratching with correct responses, as it could be some form of anticipation, positive arousal of receiving food rewards, or a possible contrast effect due to the ratio of correct and incorrect responses.

## 5. Conclusions

In conclusion, despite having a reputation for being less emotionally responsive than chimpanzees, bonobos also show higher rates of SDBs in response to emotional arousal. Whereas self-scratching appears the most common SDB in chimpanzees, the bonobos in this study most commonly performed nose wipes. The fact that nose wipes were so common among the bonobos, and are potentially indicative of low levels of arousal, could hint to a species-specific difference in emotional reactivity. Although more research is necessary to better comprehend these expressive patterns of SDBs and their reliability as indicators of emotional arousal in bonobos, we were able to provide evidence that some, but not all SDBs, increase with putative negative arousal, namely nose wiping and rough self-scratching. Arousal-related hemispace effects for rough self-scratching provide further reason to believe that this behaviour may indicate negative arousal. In contrast, we found that gentle self-scratching increased with possible positive arousal. Overall, we encourage future studies to investigate SDBs while taking into account the nuances laid out in this study.

## Figures and Tables

**Figure 1 animals-12-03002-f001:**
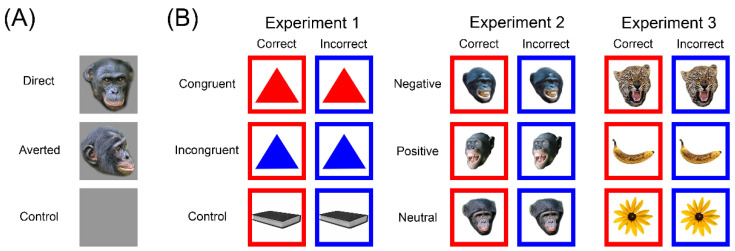
Examples of stimuli used in (**A**) Study 1, the response-slowing task and (**B**) Study 2, the pictorial emotional Stroop task.

**Figure 2 animals-12-03002-f002:**
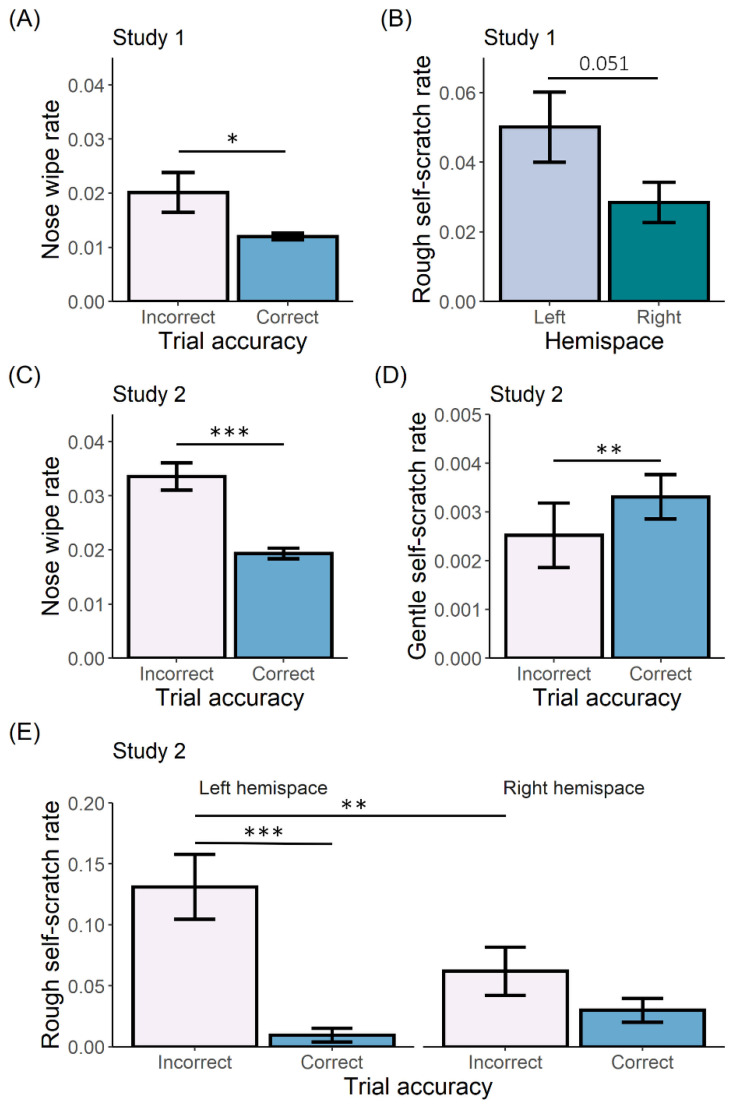
Mean rates per trial per second of (**A**) nose wiping in Study 1 in function of trial accuracy; (**B**) rough self-scratching in Study 1 in function of hemispace; (**C**) nose wiping in Study 2 in function if trial accuracy; (**D**) gentle self-scratching in Study 2 in function of trial accuracy; and (**E**) rough self-scratching in Study 2 in function of trial accuracy and hemispace *** *p* < 0.001; ** *p* < 0.01; * *p* < 0.05.

**Table 1 animals-12-03002-t001:** Subject information of the eight bonobos included in this study.

Subject	Sex	Age	Study 1	Study 2
Busira	Female	16	Yes	Yes
Habari	Male	14	Yes	Yes
Kianga	Female	17	Yes	No
Kikongo	Male	7	Yes	No
Mokonzi	Male	7	Yes	Yes
Nayembi	Female	15	Yes	No
Vifijo	Male	27	Yes	No
Zamba	Male	22	Yes	Yes

**Table 2 animals-12-03002-t002:** Occurrence of the different SDB types per Study.

	Nose Wipe	Gentle Scratch	Rough Scratch	Self-Touch
Study 1 (RST)	69.2%	19.8%	9.1%	1.9%
Study 2 (MEST)	80.6%	11.9%	5.6%	1.9%
Total (% of total)	75.5%	18.4%	7.2%	1.9%

**Table 3 animals-12-03002-t003:** Individual Handedness- and Side-indices across tasks and SDB types. Please note that the HI indices for the different SDBs are influenced by HI-screen.

Subject	HI-Tube	HI-Screen	HI-Gentle Scratch	HI-Rough Scratch	HI-Nose Wipe	SI-Gentle Scratch	SI-Rough Scratch
Busira	−0.01	0.92	0.79	0.77	0.98	0.28	−0.15
Habari	0.61	−0.47	−0.33	−0.69	−0.35	0.25	0.43
Kianga	−0.76	0.00	−0.21	−0.47	0.16	−0.32	0.20
Kikongo	0.15	0.96	−0.27	NA	−0.54	0.45	NA
Mokonzi	0.25	0.52	0.03	−0.50	−0.03	0.10	0.5
Nayembi	1.00	0.72	0.33	0.40	1.00	0.33	−0.20
Vifijo	0.72	0.97	1.00	0.33	0.78	0.00	−0.33
Zamba	0.68	−0.67	0.00	0.41	−0.17	−0.50	−0.76

**Table 4 animals-12-03002-t004:** Results of the LMM examining the interaction between trial accuracy and hemispace on the rate of SDBs.

Study 1				Study 2			
*Nose wipe*	Chisq	df	*p*	*Nose wipe*	Chisq	df	*p*
Accuracy	5.989	1	**0.014**	Accuracy	21.216	1	**<0.001**
*Gentle Scratch*	Chisq	df	*p*	*Gentle Scratch*	Chisq	df	*p*
Accuracy ^a^	0.277	1	0.599	Accuracy ^a^	8.328	1	**0.004**
Hemispace ^a^	0.558	1	0.455	Hemispace ^a^	0.001	1	0.969
Accuracy * Hemispace	0.102	1	0.750	Accuracy * Hemispace	0.031	1	0.860
*Rough scratch*	Chisq	df	*p*	*Rough scratch*	Chisq	df	*p*
Accuracy ^a^	0.816	1	0.366	Accuracy	15.604	1	**<0.001**
Hemispace ^a^	3.866	1	**0.049**	Hemispace	0.086	1	0.769
Accuracy * Hemispace	0.146	1	0.702	Accuracy * Hemispace	6.469	1	**0.011**

^a^ Results are from models in which the non-significant interaction effect was removed. Bold values highlight significant results. * the interaction effect of the inputted fixed factors.

## Data Availability

The data that support the findings of this study are available from the corresponding author upon reasonable request.

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
