# Peer review of "Evaluating Self-Directed Behaviours and Their Association with Emotional Arousal across Two Cognitive Tasks in Bonobos (Pan paniscus)"

_animals, 2022, doi:10.3390/ani12213002_

Round 1

Reviewer 1 Report

In general, this is a clear and well-written paper that focuses on handedness of particular self-directed behaviors in bonobos.  The introduction clearly explains the context for this research, and the methods are straightforward, generally clear, and appropriate.  data

The wording used to describe some of the statistics should be changed.  E.g., in ll. 237-239, the authors state that the hand used for using the touchscreen ‘influenced subsequent hand use for performing SDBs.  It is true that these two things are not independent of one another, but not the case that the first influenced the second.

Most of my comments address clarifying the writing:

ll. 55-56.  Increased rates of SDB’s?

l. 62.  ‘SDBs also’ indicate?  reveal?  ‘Display’ is not the best word choice here.

l. 66.  self-grooming or self-plucking.  Either put a comma before ‘such’ or delete the comma after self-plucking.

l.84.  To what does ‘it’ refer here?

ll. 87-89  Unclear.  I am not sure what this means. 

ll. 100-101.  Also unclear. 

l. 105.  As an indicator: insert ‘an’ between ‘as’ and ‘indicator.’

l. 108 – “Bonobos represent an interesting case to study frequency and asymmetry of SDBs during cognitive tasks.”  Clarify the next few lines by stating where this is going.  E.g. Bonobos are an excellent study species for an examination of frequency and asymmetry of SDBs during cognitive tasks for several reasons.  First, chimpanzees and bonobos appear to differ in handedness. [here give your examples]. Second, bonobos and chimpanzees may differ in their emotional decision making…..

l. 119.  Use ‘In addition’ or ‘Further’ in place of ‘on the other hand.’

ll. 162-63 Replace ‘was extended until’ with ‘was increased to.’

l. 171 Replace completed with ‘conducted.’

l. 173.  Replace ‘comprised’ with ‘were.’

l. 177 Replace ‘is’ with ‘was.’

l. 183-888.  I am not sure that I understand this section.  Is the below correct?  If not, please reword for clarity.

The study consisted of three parts: 

1) a colour-interference Stroop test in which bonobos were shown photographs of geometric shapes that were either the same colour as the frame or a different color; 

2) a social, pictorial, emotional Stroop test in which bonobos were shown images of unfamiliar bonobos that had different facial expressions and that were in negative, neutral or positive contexts;

3) a non-social, pictorial emotion Stroop test in which bonobos were shown biologically relevant valent objects that were predicted to have a negative, neutral, or positive association.

l. 194.  What is the ‘accuracy’ of the trial?

l. 252. Remove the comma after the parentheses

l. 451.  Should be ‘in an attempt’ here.

ll. 465-466.  A word appears to be missing here.  …’could hint to a species-specific ?? in emotional reactivity.’

l. 224.  (82) 90.6%) of trials.  (i.e. insert the word ‘of’ here for clarity)

l. 346.  Remove the second sentence in the Discussion, which is redundant with material in the Introduction.

In general, this is a clear and well-written paper that focuses on handedness of particular self-directed behaviors in bonobos, an interesting topic.  The introduction clearly explains the context for this research, and the methods are straightforward, generally clear, and appropriate.

The statistics are well done but the wording used to describe some of the statistics should be changed.  E.g., in ll. 237-239, the authors state that the hand used for using the touchscreen ‘influenced subsequent hand use for performing SDBs.  It is true that these two things are not independent of one another, but not the case that the first influenced the second.

Most of my comments address clarifying the writing:

ll. 55-56.  Increased rates of SDB’s?

l. 62.  ‘SDBs also’ indicate?  reveal?  ‘Display’ is not the best word choice here.

l. 66.  self-grooming or self-plucking.  Either put a comma before ‘such’ or delete the comma after self-plucking.

l.84.  To what does ‘it’ refer here?

ll. 87-89  Unclear.  I am not sure what this means. 

ll. 100-101.  Also unclear. 

l. 105.  As an indicator: insert ‘an’ between ‘as’ and ‘indicator.’

l. 108 – “Bonobos represent an interesting case to study frequency and asymmetry of SDBs during cognitive tasks.”  Clarify the next few lines by stating where this is going.  E.g. Bonobos are an excellent study species for an examination of frequency and asymmetry of SDBs during cognitive tasks for several reasons.  First, chimpanzees and bonobos appear to differ in handedness. [here give your examples]. Second, bonobos and chimpanzees may differ in their emotional decision making…..

l. 119.  Use ‘In addition’ or ‘Further’ in place of ‘on the other hand.’

ll. 162-63 Replace ‘was extended until’ with ‘was increased to.’

l. 171 Replace completed with ‘conducted.’

l. 173.  Replace ‘comprised’ with ‘were.’

l. 177 Replace ‘is’ with ‘was.’

l. 183-888.  I am not sure that I understand this section.  Is the below correct?  If not, please reword for clarity.

The study consisted of three parts: 

1) a colour-interference Stroop test in which bonobos were shown photographs of geometric shapes that were either the same colour as the frame or a different color; 

2) a social, pictorial, emotional Stroop test in which bonobos were shown images of unfamiliar bonobos that had different facial expressions and that were in negative, neutral or positive contexts;

3) a non-social, pictorial emotion Stroop test in which bonobos were shown biologically relevant valent objects that were predicted to have a negative, neutral, or positive association.

l. 194.  What is the ‘accuracy’ of the trial?

l. 252. Remove the comma after the parentheses

l. 451.  Should be ‘in an attempt’ here.

ll. 465-466.  A word appears to be missing here.  …’could hint to a species-specific ?? in emotional reactivity.’

l. 224.  (82) 90.6%) of trials.  (i.e. insert the word ‘of’ here for clarity)

l. 346.  Remove the second sentence in the Discussion, which is redundant with material in the Introduction.

Reviewer 2 Report

This is a clearly described investigation of self-directed behaviour in captive bonobos. The animals participated voluntarily and the behaviour was investigated in two different tests. Results are based on a small number of subjects but a large number of behavioural events. The findings could be indicative of lateralised behavioural responses to arousal and frustration. I found the article intriguing and well-written, and have only a few requests.

2.2. Testing procedure – Please explain the timing of Study 1 and 2. Over what timespan was each study conducted and what was the interval between them?

Given that there were 20 individuals in the group and all could potentially choose to participate, why did only 8, and subsequently 4, participate in the research? Did you exclude participants that did not perform high levels of the recorded behaviours?

Am I correct in understanding that any of the 20 individuals could freely enter and leave the test room freely throughout the data collection periods? How long were the sessions involving individual animals? Did you limit the length? Did you take steps to exclude individuals who monopolised the touch screen so that others could have opportunities to participate?

2.3. Touchscreen tasks – Please provide a figure showing the stimuli used in the two tests. How did you determine which stimuli would be rewarded?

2.4. Video coding  - Please provide short video clips illustrating the four behaviours.

2.6. Data analyses - Please indicate the inter-observer reliability of the behavioural scoring.

Results – It would be of interest to know if the timing relative to the fission-fusion events affected the behaviours in Study 1, and whether the behaviours were affected by which stimuli types were shown (e.g. direct vs indirect gaze, valence of facial expression and objects). 

Discussion – Please comment on the possibility that some individuals were repeatedly performing certain self-directed behaviours as stereotypies (abnormal repetitive behaviour). Also, given that other individuals could be present during the testing, please comments on the possibility that some of the behavioural responses could have been socially facilitated by the performance of the same behaviours by bystanders.

Reviewer 3 Report

See attched file please

Reviewer 4 Report

I found this to be an interesting and scientifically sound study exploring several specific but important aspects of self-directed behaviours in captive bonobos. The results of the study are important to improve our understanding of the reliability of self-directed behaviours as indicators of emotional arousal in bonobos and, more generally, to improve the precision with which we encode the behaviour of this species. My only significant comments are related to the “valence hypothesis” explaining lateralization of emotional processing. From my point of view, the study has too little to do with this topic so it would be better to avoid this discussion completely. Alternatively, more precise and in-depth analysis of this hypothesis is needed to make this discussion meaningful.

L 40-41: The Abstract in general is clear and fluent. I believe however that without any previous explanation the sentence about the “valence hypothesis” is not informative for a reader. I recommend skipping it here.

L 86-87: The valence hypothesis in its full form postulates that one hemisphere is specialized in one valence of emotions and the other hemisphere – on the other. It would be better to explain it here to make the name of the hypothesis clear to a reader. Please see also this review Najt, P., Bayer, U., & Hausmann, M. (2013). Models of hemispheric specialization in facial emotion perception—a reevaluation. Emotion, 13, 159–167. doi:10.1037/a0029723

L 87-89: Here you mention the “modified valence hypothesis” but do not actually explain what it is. As a result, the information appears to be not useful for understanding your study. Moreover, you never mention this modified hypothesis again throughout the MS which means this information is not necessary. It would be better to skip it (or at least provide more details).

L389-392: The conclusion that your results support the ‘valence hypothesis’ is very confusing to me. You state that your results “support the ‘valence hypothesis’ proposing that the right hemisphere is dominant for processing arousal”. You state in the Introduction that: “According to the ‘right- hemisphere hypothesis’, both negative and positive emotions are mainly processed in the right brain hemisphere [28,29] while the ‘valence hypothesis’ proposed that the right hemisphere is more specialized in processing (only) negative emotions [30,31].”. That is, the result that rough self-scratches were more directed to the left hemispace when bonobos were aroused fits the predictions of both hypotheses (and does not support or contradict one of them)! As more clearly explained in Najt, Bayer, & Hausmann (2013) the ‘valence hypothesis’ proposes oppositely directed lateralizations for positive and negative emotions. Left-sided bias associated with the arousal per se doesn’t support this hypothesis. It doesn’t make your work less significant as you investigate other important questions. The necessity of the discussion of the ‘valence hypothesis’ in the context of your study seems questionable.

L 411-413: How is it possible to say that the individual has a dominant right brain hemisphere? In light of the current knowledge, we usually talk about hemispheric dominance for a specific function or task. The idea of complete dominance of one hemisphere in a particular individual can be now called outdated. The dominance of one or another brain hemisphere depends on the specific cerebral functions involved in the particular behaviour. This is a widely accepted concept (see e.g., Rogers, L. J. (2002). Lateralization in vertebrates: its early evolution, general pattern, and development. In Advances in the Study of Behavior (Vol. 31, pp. 107-161). Academic Press; Vallortigara, G., & Rogers, L. (2005). Survival with an asymmetrical brain: advantages and disadvantages of cerebral lateralization. Behavioral and brain sciences; MacNeilage, P. F., Rogers, L. J., & Vallortigara, G. (2009). Origins of the left & right brain. Scientific American301(1), 60-67.

L 438: How straightforward was the differentiation between gentle and rough scratching? Maybe I missed this information in the Methods but if not and it  is not clearly described, please add this. Considering the differences you found, it should be clear to the reader and future investigators how distinguishable these behaviours are.
